# Cemiplimab for Locally Advanced and Metastatic Cutaneous Squamous-Cell Carcinomas: Real-Life Experience from the French CAREPI Study Group

**DOI:** 10.3390/cancers13143547

**Published:** 2021-07-15

**Authors:** Candice Hober, Lisa Fredeau, Anne Pham-Ledard, Marouane Boubaya, Florian Herms, Philippe Celerier, François Aubin, Nathalie Beneton, Monica Dinulescu, Arnaud Jannic, Nicolas Meyer, Anne-Bénédicte Duval-Modeste, Laure Cesaire, Ève-Marie Neidhardt, Élodie Archier, Brigitte Dréno, Candice Lesage, Clémence Berthin, Nora Kramkimel, Florent Grange, Julie de Quatrebarbes, Pierre-Emmanuel Stoebner, Nicolas Poulalhon, Jean-Philippe Arnault, Safia Abed, Bertille Bonniaud, Sophie Darras, Valentine Heidelberger, Suzanne Devaux, Marie Moncourier, Laurent Misery, Sandrine Mansard, Maxime Etienne, Florence Brunet-Possenti, Caroline Jacobzone, Romain Lesbazeilles, François Skowron, Julia Sanchez, Stéphanie Catala, Mahtab Samimi, Youssef Tazi, Dominique Spaeth, Caroline Gaudy-Marqueste, Olivier Collard, Raoul Triller, Marc Pracht, Marc Dumas, Lucie Peuvrel, Pierre Combe, Olivier Lauche, Pierre Guillet, Yves Reguerre, Ingrid Kupfer-Bessaguet, David Solub, Amélie Schoeffler, Christophe Bedane, Gaëlle Quéreux, Sophie Dalac, Laurent Mortier, Ève Maubec

**Affiliations:** 1Centre Hospitalier Universitaire (CHU) de Lille, 59037 Lille, France; candice.hober.etu@univ-lille.fr (C.H.); Laurent.MORTIER@CHRU-LILLE.FR (L.M.); 2Hôpital Avicenne, Assistance Publique–Hôpitaux de Paris (APHP), 93000 Bobigny, France; lisa.fredeau@aphp.fr (L.F.); marouane.boubaya@aphp.fr (M.B.); 3CHU de Bordeaux and University of Bordeaux, 33000 Bordeaux, France; anne.pham-ledard@chu-bordeaux.fr; 4Hôpital Saint-Louis, APHP, 75010 Paris, France; florian.herms@aphp.fr; 5CH Saint-Louis de la Rochelle, 17000 La Rochelle, France; Philippe.CELERIER@ght-atlantique17.fr; 6Université de Bourgogne–Franche-Comté and CHU de Besançon, 25000 Besançon, France; faubin@chu-besancon.fr; 7CH du Mans, 72037 Le Mans, France; nbeneton@ch-lemans.fr; 8Hôpital Pontchaillou, 35000 Rennes, France; monica.dinulescu@chu-rennes.fr; 9Hôpital Henri-Mondor, APHP, 94000 Créteil, France; arnaud.jannic@aphp.fr; 10Institut Universitaire du Cancer de Toulouse, 31100 Toulouse, France; meyer.n@chu-toulouse.fr; 11CHU de Toulouse, 31300 Toulouse, France; 12Hôpital Charles-Nicolle, 76038 Rouen, France; ab.duval-modeste@chu-rouen.fr; 13Hôpital Côte-de-Nacre, 14000 Caen, France; laure.cesaire@live.fr; 14Centre Léon-Bérard, 69008 Lyon, France; eve-marie.neidhardt@lyon.unicancer.fr; 15Hôpital Saint-Joseph, 13008 Marseille, France; earchier@hopital-saint-joseph.fr; 16CHU de Nantes and Université de Nantes, 44000 Nantes, France; brigitte.dreno@atlanmed.fr (B.D.); gaelle.quereux@chu-nantes.fr (G.Q.); 17Centre d’Investigation Clinique 1413, Institut National de la Santé et de la Recherche Médicale (INSERM), CHU de Nantes, 44000 Nantes, France; 18Centre de Recherche en Cancérologie et Immunologie Nantes Angers (CRCINA), Institut National de la Santé et de la Recherche Médicale (INSERM), 44007 Nantes, France; 19CHU de Montpellier, 34295 Montpellier, France; candice-lesage@chu-montpellier.fr; 20CHU d’Angers, 49100 Angers, France; Clemence.Berthin@chu-angers.fr; 21APHP, Hôpital Cochin, 75014 Paris, France; nora.kramkimel@aphp.fr; 22CHU de Reims, 51092 Reims, France; fgrange@ch-valence.fr (F.G.); J.SANCHEZ@ch-stquentin.fr (J.S.); 23CH de Valence, 26000 Valence, France; f.skowron@hopitaux-drome-nord.fr; 24CH Annecy Genevois, 74370 Annecy, France; jdequatrebarbes@ch-annecygenevois.fr; 25CHU de Nîmes, 30900 Nîmes, France; pierre.stoebner@chu-nimes.fr; 26UMR CNRS 5247, Université Montpellier I, 34090 Montpellier, France; 27Hôpital Lyon Sud–Hospices Civils de Lyon, 69310 Lyon, France; nicolas.poulalhon@chu-lyon.fr; 28CHU Amiens-Picardie, 80000 Amiens, France; arnault.jean-philippe@chu-amiens.fr; 29Hôpital d’Instruction des Armées Sainte-Anne, 83000 Toulon, France; safia.abed@intradef.gouv.fr; 30CHU F.-Mitterrand Dijon-Bourgogne, 21000 Dijon, France; bertille.bonniaud@chu-dijon.fr (B.B.); christophe.bedane@chu-dijon.fr (C.B.); sophie.dalac@chu-dijon.fr (S.D.); 31CH de Boulogne-sur-Mer, 62200 Boulogne-sur-Mer, France; sec_dermato@ch-boulogne.fr; 32CH Robert-Ballanger, 93600 Aulnay-sous-Bois, France; valentine.heidelberger@aphp.fr; 33CH Côte Basque, 64109 Bayonne, France; sdevaux@ch-cotebasque.fr; 34CHU de Grenoble-Alpes, 38700 Grenoble, France; mmoncourier@chu-grenoble.fr; 35CHU de Brest and University of Bretagne Occidentale, 29200 Brest, France; laurent.misery@chu-brest.fr; 36CHU de Clermont-Ferrand, 63100 Clermont-Ferrand, France; smansard@chu-clermontferrand.fr; 37CH de Cornouaille, CH Intercommunal de Quimper, 29000 Quimper, France; m.etienne@ch-cornouaille.fr; 38APHP, Hôpital Bichat, 75018 Paris, France; florence.brunet-possenti@aphp.fr; 39Hôpital du Scorff, 56322 Lorient, France; c.jacobzoneleveque@ghbs.bzh; 40CHU de Poitiers, 86021 Poitiers, France; romain.lesbazeilles@ch-niort.fr; 41CH de Niort, 79000 Niort, France; ingrid.kupfer@ch-niort.fr; 42Clinique Saint-Pierre, 66000 Perpignan, France; stepscatala.2020@gmail.com; 43CH Régional Universitaire Trousseau de Tours, 37170 Chambray les Tours, France; mahtab.samimi@univ-tours.fr; 44ISP1282 UMR INRA-Université de Tours, 37000 Tours, France; 45Clinique Sainte-Anne, 67000 Strasbourg, France; csaler@solcrr.org; 46Centre d’Oncologie de Gentilly, 54000 Nancy, France; d.spaeth@ilcgroupe.fr; 47University of Aix—Marseille and CHU de la Timone, 13005 Marseille, France; caroline.gaudy@ap-hm.fr; 48Institut de Cancérologie de la Loire–Lucien-Neuwirth (ICLN), 42270 Saint-Priest-en-Jarez, France; olivier.collard@icloire.fr; 49Institut Franco-Britannique, 92300 Levallois-Perret, France; rtriller@orange.fr; 50Groupe Hospitalier de St-Malo, 35400 St-Malo, France; m.pracht@rennes.unicancer.fr; 51CH René-Dubos, 95300 Pontoise, France; marc.dumaslattaque@aphp.fr; 52Institut de Cancérologie de l’Ouest, 44800 Saint-Herblain, France; Lucie.Peuvrel@ico.unicancer.fr; 53Pôle Santé Léonard-de-Vinci, 37170 Chambray-les-Tours, France; p.combe@cort37.fr; 54Clinique Clémentville, 34070 Montpellier, France; olivier.lauche@oncoclem.org; 55Hôpital Privé Toulon Hyères Saint-Jean, 83100 Toulon, France; pierre.guillet@clinique-st-jean.fr; 56CHU de Saint-Denis, 97400 Saint-Denis de la Réunion, France; yves.reguerre@chu-reunion.fr; 57Hôpital Louis Pasteur, 28630 Le Coudray, France; dsolub@ch-chartres.fr; 58CH Régional Metz-Thionville, 57100 Metz, France; a.schoeffler@chr-metz-thionville.fr; 59CHU de Limoges, 87000 Limoges, France; 60INSERM U 1189, University of Lille, 59037 Lille, France; 61Campus de Bobigny—Université Sorbonne Paris Nord, 93017 Bobigny, France; 62UMR 1124, Campus Saint Germain des Prés, 75006 Paris, France

**Keywords:** PD-1–blocking antibody, cemiplimab, cutaneous squamous cell carcinoma, real-life setting, immunocompromised, chronic dermatosis

## Abstract

**Simple Summary:**

Prognosis of advanced cutaneous squamous-cell carcinoma (CSCC) is poor. Recent clinical trials have shown that immunotherapy achieves significantly improved survival of patients with advanced CSCCs. However, few real-world data are available on treatment patterns and clinical outcomes of patients with advanced CSCCs receiving anti-programmed cell-death protein-1 (PD-1). To approach this issue, we conducted a retrospective study on 245 patients with advanced CSCCs from 58 centers who had been enrolled in an early-access program; 240 received cemiplimab. Our objectives were to evaluate, in the real-life setting, best overall response rate, progression-free survival, overall survival and safety. Results demonstrated cemiplimab efficacy in patients with advanced CSCCs, regardless of immune status. Patients with good Eastern Cooperative Oncology Group performance status benefited more from cemiplimab. The safety profile was acceptable.

**Abstract:**

Although cemiplimab has been approved for locally advanced (la) and metastatic (m) cutaneous squamous-cell carcinomas (CSCCs), its real-life value has not yet been demonstrated. An early-access program enrolled patients with la/mCSCCs to receive cemiplimab. Endpoints were best overall response rate (BOR), progression-free survival (PFS), overall survival (OS), duration of response (DOR) and safety. The 245 patients (mean age 77 years, 73% male, 49% prior systemic treatment, 24% immunocompromised, 27% Eastern Cooperative Oncology Group performance status (PS) ≥ 2) had laCSCCs (35%) or mCSCCs (65%). For the 240 recipients of ≥1 infusion(s), the BOR was 50.4% (complete, 21%; partial, 29%). With median follow-up at 12.6 months, median PFS was 7.9 months, and median OS and DOR were not reached. One-year OS was 73% versus 36%, respectively, for patients with PS < 2 versus ≥ 2. Multivariate analysis retained PS ≥ 2 as being associated during the first 6 months with PFS and OS. Head-and-neck location was associated with longer PFS. Immune status had no impact. Severe treatment-related adverse events occurred in 9% of the patients, including one death from toxic epidermal necrolysis. Cemiplimab real-life safety and efficacy support its use for la/mCSCCs. Patients with PS ≥ 2 benefited less from cemiplimab, but it might represent an option for immunocompromised patients.

## 1. Introduction

Cutaneous squamous-cell carcinoma (CSCC) is the second most common skin cancer after basal-cell carcinoma [1]. In Europe, the reported age-standardized CSCC incidence ranges from 15 to 77 per 100,000 individuals per year, predominantly occurring in males [2,3]. The incidence is constantly increasing, probably because of early CSCC resection, population aging and changing UV-exposure habits [4].The CSCC risk is heightened for immunocompromised patients, being about 100-times higher after organ transplantation [5,6,7,8,9], and for those CSCC oncogenic human papillomavirus-positive, or with chronic dermatitis, exposure to arsenic or ionizing radiation, or genodermatosis (e.g., dystrophic epidermolysis bullosa, xeroderma pigmentosum, albinism and Muir–Torre syndrome) [10,11,12,13,14,15]. At an early stage, CSCC prognosis is excellent, with 90% 10-year survival [16]. However, ~5% of the patients experience local recurrences, ~4% of them develop regional disease and outcomes are fatal for ~2% [16,17,18,19,20,21,22]. According to American data [23], the CSCC mortality rate is of the same order of magnitude as that of melanoma. The 5-year overall survival (OS) rate of patients with resectable, regional CSCCs was 50–60% [18,19]. The prognosis becomes more uncertain for locally advanced or metastatic disease, with either regional or distant metastases.

Since 2018, anti-programed cell-death protein-1 (PD-1) monoclonal antibodies have emerged as first-line treatments for the management of unresectable, locally advanced or metastatic CSCCs. Cemiplimab was the first immunotherapy approved by the Food and Drug Administration and the European Medicines Agency [24], followed by pembrolizumab, in the United States, for patients who are not candidates for curative radiotherapy or surgery [25]. Immunotherapies have demonstrated anti-tumor activity with response rates exceeding 40% and acceptable safety profiles [24,25,26,27].

In France, an early-access program made cemiplimab available to patients with locally advanced or metastatic CSCCs during the time between completion of enrollment in cemiplimab clinical trials and its regulatory approval. This retrospective, multicenter CAREPI trial aimed to evaluate cemiplimab efficacy and safety in the real-life setting of those early-access patients. Our results confirmed cemiplimab efficacy in real life and identified clinical characteristics of those patients associated with progression-free survival (PFS) and OS.

## 2. Materials and Methods

Patients eligible for the early-access program (August 2018 to October 2019) were adults with locally advanced or metastatic CSCCs not amenable to surgery. Exclusion criteria were active autoimmune diseases or infections, uncontrolled brain metastases, pregnancy or breastfeeding. Patients received intravenous cemiplimab infusions (3 mg/kg every 2 weeks) until death from any cause, unacceptable toxicity, or patient’s or physician’s decision. Investigators were asked to complete a standardized case-report form for each patient included in the early-access program.

This retrospective study was approved by the local Avicenne Hospital Ethics Committee (CLEA-2019-75). The national database has been declared to the French data-protection agency (CNIL approval number 2215607). In compliance with French law, consent regarding non-opposition to collect and use the data was obtained from each patient.

The primary endpoint was the best overall response rate (BOR); secondary endpoints included PFS, OS, duration of response (DOR) and safety. Standard-of-care tumor assessments were carried out at the treating facility without central review. Adverse events (AEs) were graded according to the Common Terminology Criteria for Adverse Events version 5. Efficacy and safety were assessed for all patients who received at least one cemiplimab infusion.

Patient characteristics are expressed as numbers (percentages) for discrete variables, and mean ± standard deviation or median (range) for continuous variables. Data cutoff was 19 June 2020. Median follow-up was estimated using the Kaplan–Meier reverse method. OS and PFS were defined, respectively, as the times from the first cemiplimab dose to death from any cause and until disease progression or death from any cause, whichever occurred first. DOR was defined as the time from BOR to first documentation of disease progression. OS, PFS, duration of cemiplimab treatment, and DOR were censored at the date of last information update, estimated using the Kaplan–Meier method and expressed as median (95% confidence intervals (CIs)). Prognostic factors associated with PFS and OS were identified with log-rank tests. A multivariate Cox proportional hazards regression model with a step function was used because Eastern Cooperative Oncology Group performance status (PS) violated the proportional hazards assumption. PS was determined twice (< or ≥6 months). The cumulative incidence of relapses was estimated according to type of response using competing-risk analyses and were compared with Gray’s test. All tests were two-sided, with significance set at *p* < 0.05. Analyses were computed with R statistical software V.4.0.3 (R Foundation for Statistical Computing, Vienna, Austria).

## 3. Results

### 3.1. Patients

All information concerning 245 patients, from 58 French centers, was collected. Five patients died before the first infusion and were not analyzed for efficacy and safety. Baseline (pre-cemiplimab) patient characteristics are reported for the 245 intent-to-treat patients in Table 1. Their mean age was 77 years, 73% were male, 27% had PS ≥ 2 and 24% were immunocompromised. Among the 59 immunocompromised, 64% had blood disorders, including 34% with chronic lymphocytic leukemia. Among the intent-to-treat population, CSCCs were 35% localized, 39% regional disease and 26% had distant metastases; 11% had chronic dermatitis and 3% had cutaneous ulcers. Two-thirds of CSCCs were located on the head and neck. Histopathological examination revealed 23% were poorly differentiated and 11% exhibited perineural invasion.

Regarding previous treatments (see Appendix A), 60% of intent-to-treat patients had received radiotherapy and 79% had undergone surgical excision. Moreover, about half had received systemic treatment, which was most frequently (38%) anti-epidermal growth factor receptor (EGFR) plus chemotherapy. Three-quarters received one line of systemic therapy before cemiplimab.

Cemiplimab administration lasted a median of 5.5 (95% CI 4.6–8.8) months, for a median of 10 (1–40) infusions for each per-protocol patient, with 29% (95% CI 23–36) of the patients still being treated beyond 12 months.

### 3.2. Efficacy Evaluation

Responses of the 240 assessable patients are detailed in Table 2: 21% complete responses and 29% partial responses, for a BOR of 50% (95% CI 44–57). Only 64% of responses were confirmed. The BORs did not differ according to immunocompromised versus immunocompetent status (50% versus 51%, respectively), with prior systemic treatment versus without (51% versus 50%, respectively; *p* = 0.9), or according to local, regional or distant disease (48%, 56% or 46%, respectively; *p* = 0.41). However, the BORs were lower for patients with PS ≥ 2 versus <2 (37% versus 56%, respectively; *p* = 0.01). Patients with chronic dermatitis tended to have poorer responses than those without (32% versus 52%, respectively; *p* = 0.06). The disease-control rate was 59.6% (95% CI 53.1–65.8).

The median time to complete response was 5.9 (range 1.7–13.6) months. Complete responders’ median treatment duration was 11.3 months (range 13–516 days), versus 7.5 months (range 43–595 days) for partial responders. The reasons for cemiplimab discontinuation were not fully available for these patients. Among the 51 complete responders, only three (6%) progressed during follow-up: two progressed on cemiplimab after 318 or 471 days of treatment and one progressed 3 months after stopping cemiplimab, which had been administered for 241 days (see Appendix A). A median of 61 days of follow-up were available for 27 (53%) complete responders after cemiplimab discontinuation: only one of them relapsed. At 1 year, relapses were significantly more frequent for partial responders (53%) than complete responders (9%) (*p* = 0.007).

With global median follow-up at 12.6 months, median PFS lasted 7.9 (95% CI, 4.9–10.7) months and 1-year PFS was 38.7%; median global OS was not reached and the 1-year OS was 63.1%; and median global DOR was not reached and the 1-year DOR rate was 62.9% (Figure 1, Figure 2 and Figure 3). The 1-year PFS and OS rates did not differ according to immune status or previous systemic treatment status (*p* > 0.21). However, their durations were significantly shorter for patients with PS ≥ 2 versus PS < 2, with respective estimated percentages (95% CI) of 25.1% (15.0–41.8%) and 43.5% (36.3–52.3%) (*p* < 0.0001) for PFS, and 36% (25–52%) and 73% (66–81%) (*p* < 0.0001) for OS. The highly significant impact of PS ≥ 2 on PFS and OS was confirmed during the first 6 months, after adjustment for age, sex, chronic dermatitis, primary CSCC site and disease stage (Table 3). After 6 months, PS was no longer associated with PFS or OS. Primary head-and-neck CSCC was also associated with a better PFS.

### 3.3. Adverse Events

One-third of the patients experienced treatment-related AEs (TRAEs; Table 4), with the most common being (in decreasing order): fatigue, arthralgias/myalgias, hepatic disorders, diarrhea and pruritus. They led to treatment discontinuation for 16 (7%) patients. Twenty-two patients experienced at least one grade-3 or higher TRAE, as detailed in Table 5. They were mostly hepatic disorders and fatigue, but also renal impairment, arthralgias/myalgias, and two kidney-transplant rejections. The death of one patient from toxic epidermal necrolysis (Lyell’s syndrome) was attributed to cemiplimab. A median of 6 (range 0–70) weeks separated cemiplimab onset and the first AE. The response rates for patients with TRAEs (54.7%) and those without (47.3%) did not differ significantly (*p* = 0.45).

## 4. Discussion

This retrospective study on 240 CSCC patients confirmed cemiplimab efficacy in the real-life setting as a curative treatment for unresectable, locally advanced or metastatic disease. Patients in this series share characteristics with the 193 patients enrolled in the phase II trial evaluating cemiplimab that led to its approval [24,26,28,29]: predominantly men, older age, 29% poorly differentiated tumors [26], and mostly head-and-neck primary locations. Unlike those study participants, our population included 24% immunocompromised patients, with 16% having blood disorders (i.e., chronic lymphocytic leukemia and other hemopathies), and 27% with PS ≥ 2. Notably, in our series, 49% had received systemic treatment before starting cemiplimab, versus 34% in the phase II trial [28], 3% had a genodermatosis and 11% had an underlying chronic dermatitis, most frequently chronic wounds. The BOR herein was 50%, including 21% complete responses, which is of the same order of magnitude as in other trials evaluating anti-PD-1 [24,25,26,27,28,29].

Our results suggest that immunocompromised patients, including those with blood disorders, respond and survive as well as immunocompetent patients, meaning they apparently benefit from anti-PD-1, despite usually being excluded from trials. However, management of these patients, particularly transplant recipients, must be extremely attentive so as to avoid rejection, as highlighted by the two kidney-transplant rejections observed herein; nonetheless, they should be included in trials. Indeed, anti-PD-1 increased the risk of graft rejection and, when rejection occurred, mortality was recently estimated at 36%, with a high risk for liver-transplant recipients [30]. An ongoing trial is evaluating the safety and efficacy of cemiplimab with everolimus/sirolimus plus prednisone or without as treatment for advanced CSCCs in kidney transplantees (NCT04339062).

Our findings also support that systemic treatment-naïve patients responded as well as pretreated patients. They also showed that frail patients with poor PS responded less well. However, because more than one-third of them responded to cemiplimab, anti-PD-1 should remain the first-line systemic treatment of choice. It is now critical to identify factors predictive of response in these frail patients. Our observations indicate that patients with underlying chronic dermatitis might respond less well to cemiplimab than patients without, but that outcome remains to be confirmed by a larger study.

Remarkably, our complete responders rarely relapsed (6%), even after stopping cemiplimab. Notably, although disease progression after an objective response was observed in 21% of responders, the risk of relapse was markedly higher for partial responders than complete responders, as previously reported for melanoma patients [31,32]. Determining responders’ factors predictive of relapse and optimal treatment duration for partial responders would contribute greatly to improving their management.

Although direct comparison is impossible, for our entire population, 1-year PFS (38.7%) and OS (63.1%) were substantially lower than in Migden et al.’s phase II study [24]. Indeed, their patients’ 1-year PFS and OS ranged between 47% and 58%, and 76% and 93%, respectively, according to the different patient subgroups [28,33]. One factor that could explain this difference would be our overestimation of the response rate, attributable to either the high frequency of unconfirmed responses or the lack of independent central review. Indeed, in Migden et al.’s phase II study [26], the response rate was overestimated by investigators (53%) compared to blinded independent central reviewers (44%), as recently demonstrated by the analysis of 20 trials that had central and investigators’ BOR assessments available [34]. However, our series’ BOR was very close to the investigators’ estimated response rate in Migden et al.’s trial [26].

Another hypothesis might be that our lower-than-expected PFS and OS might reflect our patients’ characteristics, i.e., 27% with PS ≥ 2, whose 1-year OS at 36% was significantly shorter, as reported for lung cancer [35], whereas that OS rate for patients with PS < 2 reached the lower threshold of the OS estimated in the cemiplimab phase II CSCC trial [28,33]. Our best model for OS included only PS, while our best model for PFS included PS and primary head-and-neck. Although it is difficult to attribute a protective effect to head-and-neck CSCCs, it can be hypothesized that the tumor mutational burden would be increased in CSCCs located at that site, a chronically sun-exposed area, compared to other cutaneous areas not chronically sun-exposed. Because high tumor mutational burden predicts prolonged survival in patients receiving anti PD-1 [36], such a higher burden might help explain the association between longer PFS and head-and-neck site retained by our multivariate analysis. Further molecular studies are needed to confirm this hypothesis. Notably, PFS and OS did not differ according to CSCC stage, prior systemic treatment status or immune status, thereby suggesting that it would be of interest to enroll immunocompromised patients in trials evaluating anti-PD-1.

The cemiplimab-safety profile for our series was comparable with that in other studies on PD-1–blocking agents to treat CSCC [24,26,37]. Most AEs were manageable, except for 16 (7% of the patients) that necessitated cemiplimab discontinuation. One cemiplimab-related death from toxic epidermal necrolysis occurred. About 20 Stevens–Johnson syndrome/toxic epidermal necrolysis cases have been reported with other inhibitors of PD-1 or its ligand [38,39,40,41,42,43,44,45,46,47,48,49,50,51,52,53,54,55]. Toxic epidermal necrolysis is responsible for high mortality [56]. According to the American Society of Clinical Oncology guidelines, cyclosporine or intravenous immunoglobulins combined with corticosteroids should be initiated when toxic epidermal necrolysis is diagnosed [57]. Indeed, with the increasing use of immune-checkpoint inhibitors, physicians should be aware of this very rare AE. Twenty-two (9%) of our patients developed severe grade-3 or -4 TRAEs, a rate consistent with previous studies on PD-1–blocking agents [24,25,26,50]. One early-onset cemiplimab-induced grade-4 drug reaction with eosinophilia and systemic symptoms with a favorable outcome occurred in a 76-year-old woman. Considering these frail patients, the safety profile seems acceptable.

Limitations of this study are its retrospective design, the lack of central assessment of disease response, the too short follow-up that precluded accurate determinations of OS, DOR, and the long-term outcomes of responders after stopping anti-PD-1. Indeed, longer follow-up would be helpful. Moreover, PFS results may not be very accurate because assessments were made according to standard of care and may have been performed at different timepoints.

## 5. Conclusions

The results of this retrospective study confirm cemiplimab’s strong anti-tumor activity and manageable safety, meaning it should be offered to patients with unresectable, locally advanced or metastatic CSCCs. Our analysis of the characteristics of CSCC patients who received cemiplimab in the real-life setting demonstrated the poor prognosis associated with PS ≥ 2. The association between head-and-neck involvement and longer PFS requires additional molecular prognostic studies to determine whether or not that site has a protective effect on PFS for patients with locally advanced or metastatic disease. Moreover, the results of this analysis indicate that cemiplimab might be beneficial for immunocompromised patients.

## Figures and Tables

**Figure 1 cancers-13-03547-f001:**
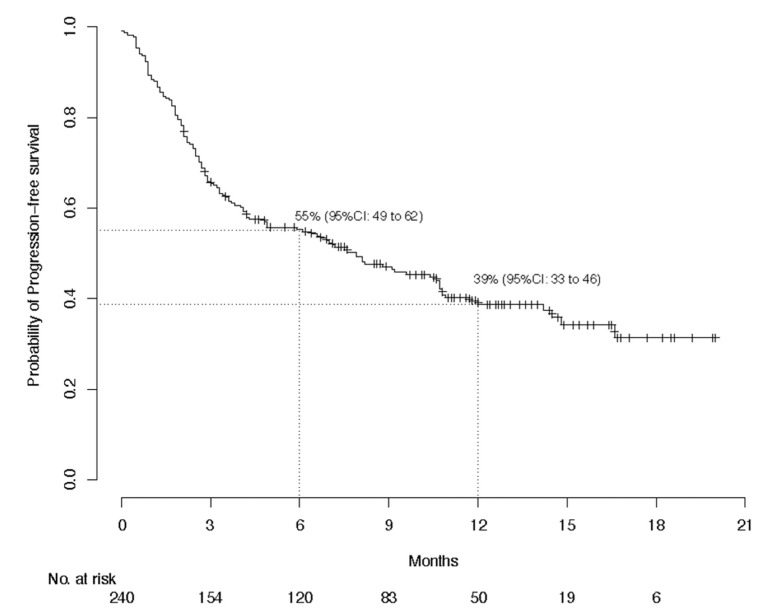
Kaplan–Meier estimations of the 6-month and 1-year probabilities of progression-free survival for the per-protocol population (*n* = 240).

**Figure 2 cancers-13-03547-f002:**
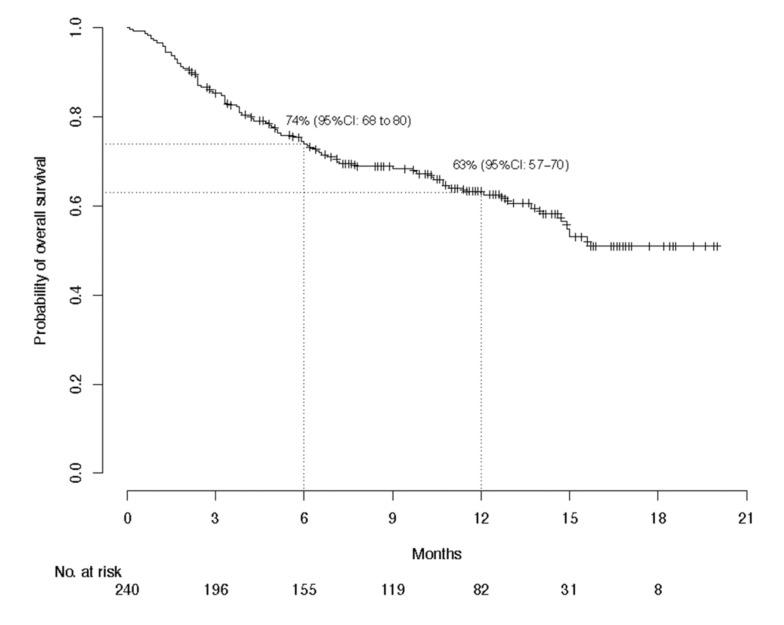
Kaplan–Meier estimations of the 6-month and 1-year probabilities of overall survival for the per-protocol population (*n* = 240).

**Figure 3 cancers-13-03547-f003:**
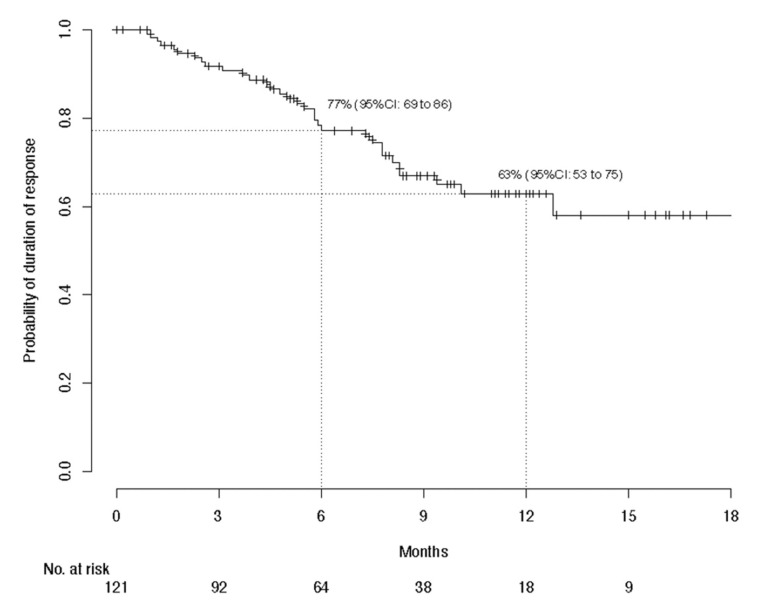
Kaplan–Meier estimations of the 6-month and 1-year probabilities of duration of response after cemiplimab treatment for the per-protocol population (*n* = 240).

**Table 1 cancers-13-03547-t001:** Baseline characteristics of all 245 intent-to-treat CSCC patients.

Characteristic	Value
Age, years	77.1 ± 13.3
Male sex	178 (73)
ECOG performance status	
0	60 (25)
1	118 (48)
≥2	66 (27)
Unknown	1 (0.4)
Immunocompromised	59 (24)
Human immunodeficiency virus-positive	8 (3)
Organ transplant	7 (3)
Chronic lymphocytic leukemia	20 (8)
Other blood disorders ^a^	18 (7)
Immunosuppressive drugs	6 (3)
Genodermatosis	8 (3)
Inherited epidermolysis bullosa	2 (0.8)
Muir–Torre syndrome	2 (0.8)
Xeroderma pigmentosum	1 (0.4)
Ichthyosis	2 (0.8)
Epidermodysplasia verruciformis	1 (0.4)
Chronic dermatitis	28 (11)
Burns	4 (1.6)
Scars	2 (0.8)
Lichen planus	2 (0.8)
Chronic wounds	9 (4)
Warts/condylomas	4 (1.6)
Arsenic keratosis	2 (0.8)
Radiodermatitis	3 (1.2)
Others ^b^	2 (0.8)
≥3 primary CSCCs	80 (33)
Primary CSCC site	
Head-and-neck ^c^	164 (70)
Trunk	9 (4)
Anorectal and/or genital	12 (5)
Arm or leg	58 (24)
Unknown	3 (1.2)
Histopathological characteristics	
Poor differentiation	57 (23)
Perineural invasion	26 (11)
Both	9 (4)
None	69 (28)
Unknown	84 (34)
CSCC stage	
Localized	85 (35)
Regional	95 (39)
Distant metastases	64 (26)
Unknown	1 (0.4)

Results are expressed as mean ± standard deviation or number (%). ECOG, Eastern Cooperative Oncology Group; CSCC, cutaneous squamous-cell carcinoma. ^a^ Other blood disorders included: polycythemia vera, four; Waldenström’s macroglobulinemia, three; two each: mantle-cell lymphoma or myelodysplastic syndrome; one each: large B-cell lymphoma, cutaneous T-cell lymphoma, essential thrombocythemia, multiple myeloma associated with amyloid light-chain amyloidosis, IgM monoclonal gammopathy, thrombopenia of unspecified cause or idiopathic CD4 lymphocytopenia. ^b^ Carcinomas due to phototherapy or erosive pustular dermatosis of the scalp. ^c^ Including two CSCCs located on the lips.

**Table 2 cancers-13-03547-t002:** Best overall responses (*n* = 240) as assessed by investigators.

Outcome	*n* (%)
Complete response	51 (21)
Confirmed	36 (15)
Unconfirmed	15 (6)
Partial response	70 (29)
Confirmed	41 (17)
Unconfirmed	29 (12)
Stable disease	22 (9)
Progressive disease	84 (35)
Not assessable	13 (5)
Best overall response rate, *n* (% [95% CI])	121 (50.4 [43.9–56.9])
Confirmed	77 (32)
Unconfirmed	44 (18)
Best overall disease control rate, *n* (% [95% CI])	143 (59.6 [53.1–65.8])

Results are expressed as number (%), unless stated otherwise.

**Table 3 cancers-13-03547-t003:** Factors associated with progression-free survival or overall survival in univariate and multivariate analyses.

Factor	Univariate		Multivariate	
	HR (95% CI)	*p*	HR (95% CI)	*p*
**Progression-free survival**				
Age	1.00 (0.98–1.01)	0.62	1.00 (0.98–1.01)	0.63
Male sex	0.79 (0.55–1.15)	0.22	0.91 (0.61–1.37)	0.66
Immunocompromised	1.03 (0.7–1.51)	0.89	1.15 (0.76–1.76)	0.5
ECOG PS ≥ 2				
≤6 months	2.3 (1.53–3.44)	<0.0001	2.33 (1.52–3.55)	0.0001
6 months	0.88 (0.31–2.51)	0.81	0.85 (0.3–2.46)	0.77
Chronic dermatitis	1.67 (1.02–2.71)	0.04	1.07 (0.61–1.87)	0.8
Primary head-or-neck CSCC	0.58 (0.41–0.81)	0.0002	0.52 (0.34–0.79)	0.0025
Localized disease	1.16 (0.82–1.64)	0.41	0.72 (0.49–1.05)	0.09
Previous systemic treatment	0.88 (0.62–1.23)	0.44	1.03 (0.71–1.50)	0.88
**Overall survival**				
Age	1.00 (0.99–1.02)	0.81	0.99 (0.98–1.01)	0.46
Male sex	0.9 (0.56–1.44)	0.66	1.01 (0.61–1.67)	0.97
Immunocompromised	0.82 (0.49–1.35)	0.43	0.91 (0.53–1.56)	0.72
ECOG PS ≥ 2				
≤6 months	4.39 (2.62–7.33)	<0.0001	4.56 (2.64–7.85)	0.0001
>6 months	1.61 (0.61–4.27)	0.34	1.69 (0.63–4.52)	0.3
Chronic dermatitis	0.98 (0.49–1.95)	0.95	0.7 (0.32–1.51)	0.36
Primary head-or-neck CSCC	0.76 (0.49–1.18)	0.22	0.67 (0.4–1.13)	0.13
Localized disease	1.02 (0.66–1.58)	0.94	0.74 (0.45–1.2)	0.22
Previous systemic treatment	0.76 (0.5–1.17)	0.21	1.09 (0.68–1.76)	0.72

ECOG PS, Eastern Cooperative Oncology Group performance status; CSCC, cutaneous squamous-cell carcinoma.

**Table 4 cancers-13-03547-t004:** Each cemiplimab-related adverse event occurred in at least two of the 240 treated patients.

Adverse Event	Any Grade	Grade ≥ 3
Any	75 (31)	22 (9)
Led to cemiplimab discontinuation	16 (7)	12 (5)
Fatigue	21 (9)	4 (2)
Arthralgias/myalgias	17 (7)	2 (1)
Cholestasis/cytolysis/hepatitis	10 (4)	5 (2)
Diarrhea	7 (3)	0
Pruritus	6 (3)	0
Rash	5 (2)	0
Hypothyroidism	5 (2)	0
Renal failure	5 (2)	3 (1)
Hyperthyroidism	4 (2)	0
Lymphopenia	3 (1)	0
Decreased appetite	3 (1)	1 (0.4)
Peripheral neuropathy	3 (1)	0
Anemia	2 (1)	0
Neutropenia	2 (1)	0
Myocarditis	2 (1)	1 (0.4)
Corticotropic insufficiency	2 (1)	0
Colitis	2 (1)	2 (1)
Vomiting	2 (1)	1 (0.4)
Loss of weight	2 (1)	0
Balance disorder	2 (1)	0
Transplant rejection	2 (1)	2 (1)

Results are expressed as number (%).

**Table 5 cancers-13-03547-t005:** Serious cemiplimab-related adverse events in the 240 treated patients.

Adverse Event	Severity Grade			
	Any	Grade 3	Grade 4	Grade 5
Cholestasis/cytolysis/hepatitis	5 (2)	3	2	0
Fatigue	4 (2)	4	0	0
Renal impairment	3 (1)	2	1	0
Arthralgias/myalgias	2 (1)	2	0	0
Colitis	2 (1)	2	0	0
Transplant rejection	2 (1)	1	1	0
Decreased appetite	1 (0.4)	1	0	0
Myocarditis	1 (0.4)	1	0	0
Vomiting	1 (0.4)	1	0	0
Acute pancreatitis	1 (0.4)	1	0	0
Interstitial lung disease	1 (0.4)	1	0	0
Drug reaction with eosinophilia and systemic symptoms	1 (0.4)	0	1	0
Toxic epidermal necrolysis	1 (0.4)	0	0	1

Results are expressed as number (%) or number.

## Data Availability

Relevant data supporting the findings of this study are available within the article and Appendix A and are available from the authors upon reasonable request.

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
