# Peer review of "Cemiplimab for Locally Advanced and Metastatic Cutaneous Squamous-Cell Carcinomas: Real-Life Experience from the French CAREPI Study Group"

_cancers, 2021, doi:10.3390/cancers13143547_

Round 1
Reviewer 1 Report
Important contribution to the literature with data on ECOG PS > 1 and immunocompromised patients, with the limitations of the study design and limited follow-up.
Specific comments:
Summary – poor English, can be improved
Abstract – also English wording could be improved.
Intro – incorrect to state that patients with regional or distant metastases have a 4% mortality rate
Limitations – should add that PFS results may not be very accurate as follow up imaging was according standard of care, so may have been done at different timepoints
Conclusion – wording needs to be modified – not correct to state that there was a a protective effect of primary on a head-or-neck site for PFS
Ref 28 and 26 are the same. 28 is the abstract from conference presentation so is not needed as 26 is the corresponding manuscript .
Reviewer 2 Report
Many thanks for the opportunity of reviewing this paper. This is an outstanding, well-written and a worthy manucript. I only have some minor points.
- When refering to neurotropism, do the authors refer to PNI…? IF this was the case, specifying the nerve diameter <0.1 or ≥0.1mm would be desirable.
- It would be interesting to have the stage at diagnosis and a more detailed information on the clinical and histopathological features of the tumors. Nevertheless, it may be difficult to obtain in such a multicenter setting and thus I consider this optional in this context. It is interesting that there were no differences in respone to treatment in patients naïve versus previously treated, and depending on the CSCC stage (local/regional/distant).
- In line 266 (27 to 28%... is this an error..?)
- In line 312 and 315, please change CSSC by CSCC
- Concerning the protective effect detected in H&N located CSCCs, the authors should compare their prognostic features with that of tumors located out of the H&N or should comment that it is difficult to attribute a protective effect of a CSCC located in the H&N area in terms of anti-PD1 response. There may be other features, more common in H&N tumors in this cohort that might impact on cemiplimab response. In the models, there are not much tumor features beyond tumor location.
- Do the authors found a better response in patients with TRAEs…? They may include it in the models in case they found it is associated with better response.
I would have been great to see waterfall and spider plots to see how responses were graphically.
